# Small Bowel Neuroendocrine Tumors—10-Year Experience of the Ottawa Hospital (TOH)

Abdulhameed Alfagih [1,2], Abdulaziz AlJassim [1,3], Nasser Alqahtani [1,4], Michael Vickers [1], Rachel Goodwin [1] and Timothy Asmis [1,*]

1   Division of Medical Oncology, Department of Medicine, The Ottawa Hospital, The University of Ottawa, Ottawa, ON K1N 6N5, Canada; aalfaqih@kfmc.med.sa (A.A.); azaljassim@moh.gov.kw (A.A.); alqahtanina3@mngha.med.sa (N.A.); mvickers@toh.ca (M.V.); rgoodwin@toh.ca (R.G.)
2   Medical Oncology Department, Comprehensive Cancer Center, King Fahad Medical City, Riyadh 11525, Saudi Arabia
3   Kuwait Cancer Control Center, Kuwait City 42262, Kuwait
4   King Abdulaziz Hospital, Ministry of National Guard Health Affairs, Al Ahsa 11426, Saudi Arabia
*   Correspondence: tiasmis@toh.ca

**Abstract:** (1) Aim: The prevalence and incidence of small bowel NETs have increased significantly over the past two decades. This study aims to report the 10-year experience of SB-NET management at a regional cancer center in Canada. (2) Materials and methods: We conducted a retrospective study of the clinical and pathological data of patients diagnosed with biopsy-proven SB-NET at The Ottawa Hospital (TOH), Ottawa, Canada between 2011 and 2021. We report the clinicopathological characteristics of these patients, as well as their outcomes data, including survival rates. (3) Results: Between 2011 and 2021, a total of 177 SB-NET cases were identified with 51% ($n = 91$) of cases being males. The most common sites of the tumors were the ileum 53% ($n = 94$), followed by the duodenum 9% ($n = 16$) and jejunum 7% ($n = 12$). Approximately 24% ($n = 42$) of the patients had symptoms for over six months prior to diagnosis and 18% ($n = 32$) had functioning SB-NET during the course of the disease. The majority of patients had locally advanced or metastatic disease at the time of presentation with stage III, and stage IV representing 42% ($n = 75$), and 41% ($n = 73$) respectively. The majority of patients 84% ($n = 148$) had well-differentiated histology. One hundred twenty patients underwent surgical resection of the primary tumor including 28 patients (16%) with limited metastatic disease. A total of 21 patients (18%) had recurrence after curative surgery. A total of 62 patients (35%) received first-line somatostatin analog (SSA) therapy for unresectable disease and seven patients had PRRT after progression on SSA. Five years OS was 100%, 91%, 97%, and 73% for stages I, II, III, and IV respectively. In univariate analysis, carcinoid symptoms, T stage, and differentiation were significant predictors for worse overall survival, but not RFS. (4) Conclusions: Compared to published historical controls, our study suggests improvement in the 5-year survival rate of SB-NETs over the last 10 years.

**Keywords:** neuroendocrine tumor; chemotherapy; PRRT; somatostatin analogues

## Brief Description

The aim of this study was to report the 10-year experience of small bowel neuroendocrine tumor (SB-NET) management at the Ottawa Hospital in Ottawa, Canada. We retrospectively analyzed 177 patients with biopsy-proven SB-NETs between 2011 and 2021. The 5-year overall survival rate was 100%, 91%, 97%, and 73% for stages I, II, III, and IV respectively. Our 10-year experience provides important information regarding the clinical features and outcomes of patients with SB-NETs.

## 1. Introduction

The incidence and prevalence of small bowel neuroendocrine tumors (SB-NETs) are increasing. This group of tumors accounts for approximately 40% of all primary small

bowel malignancies. SB-NETs arise from the neuroendocrine secretory cells, which can produce peptides causing hormonal syndromes (carcinoid syndrome) [1–3]. Due to the advancements in the understanding of the biology of NETs, it has become possible to identify its various molecular alterations based on advanced molecular studies and next-generation sequencing. The current WHO classification of neuroendocrine neoplasms (NEN), established in a consensus meeting in Lyon, divides them into NETs and neuroendocrine carcinomas (NEC) based on their histological features, growth rate, and molecular differences. NECs usually have TP53 or RB1 mutations, while well-differentiated NETs typically have MEN1, DAXX, and ATRX mutations. These distinct genetic profiles contribute to the varying behavior and clinical outcomes observed in NECs and well-differentiated NETs. NECs have a worse prognosis than well-differentiated NETs, and the treatment for NECs is typically more aggressive [4–6]. Previously, mixed NENs were classified based on the predominant cell type. However, genomic data have shown that these tumors often have a mixed genetic profile, with features of both NETs and non-neuroendocrine tumors. As a result, mixed NENs are now grouped into a new conceptual category called "mixed neuroendocrine–non-neuroendocrine neoplasms (MiNENs)". This category includes tumors that have both NET and non-NET features, as well as tumors that have features of both adenocarcinomas and NECs.

NETs are characterized by their heterogeneous histologies and are often associated with various clinical courses. They can also be classified according to their grade, histologic differentiation, and location of the primary NET. The World Health Organization has introduced a new classification system that categorizes NETs into groups based on their histologic differentiation and Ki67% proliferative index. Well-differentiated NETs are characterized by a variety of cells that have a semi-solid appearance and a round nucleus [4]. Up to 30% of NETs are characterized by clinical symptoms that are related to the secretion of hormones [7]. Ref. [4] SB-NETs are difficult to diagnose because of their nonspecific symptoms and hard-to-reach location. The diagnosis is based on clinical symptoms, biochemical tests, radiological imaging, and endoscopic data. Somatostatin receptor-based imaging and endoscopic procedures, such as video capsule endoscopy (CE) and double-balloon enteroscopy (DBE), have improved the diagnosis of SB-NETs [8].

Surgical resection remains the gold standard management for localized SB-NET. Macroscopic radical resection not only reduces the risk of bowel obstruction and ischemia-related complications but also leads to improved outcomes. It is recommended to combine macroscopic radical resection with mesenteric lymphadenectomy for optimal results [9,10]. A recent study by S. Levy et al. demonstrated that patients who undergo primary tumor resection in the setting of stage IV disease experience a significant improvement in disease-specific mortality [11,12]. Liver metastasectomy has shown promise as an effective treatment for neuroendocrine tumor (NET) liver metastases. Nevertheless, the decision to undergo surgery should be made on an individual basis, considering the unique characteristics of each patient and after a thorough selection process [13].

Liver transplantation can be a viable treatment option for patients with NET liver metastases who have exhausted other treatment options. The success of liver transplantation for NET liver metastases is relatively high. Some studies have reported 5-year survival rates of up to 50% [14,15]. Patients having NET liver metastases that are not amenable to surgery may benefit from liver-directed therapies such as thermal ablation, radiofrequency ablation, transarterial chemoembolization (TACE), selective internal radiotherapy with yttrium-90 ($^{90}$Y)-microspheres, and hepatic intra-arterial injection of $^{90}$Y-DOTA-lanreotide [16–18].

Over the last decades, several breakthroughs redirected the approach to the management of advanced NETs [19]. The use of somatostatin analogs (SSAs) to treat patients with NETs has been established as the first-line therapy. SSAs are typically used as first-line therapy in most patients with unresectable/metastatic midgut NETs with the aim of controlling carcinoid syndrome and tumor growth. Octreotide LAR was evaluated in PROMID Study in comparison to placebo in patients with midgut NETs and showed a significantly

longer time to tumor progression both in functioning and non-functioning tumors, with a median time to tumor progression of 14 vs 6 months, for octreotide LAR vs placebo groups respectively [20]. Lanreotide was evaluated in the CLARINET study in comparison with placebo in patients with advanced, well-differentiated or moderately differentiated, nonfunctioning, somatostatin receptor–positive NETs of grade 1 or 2, originating from midgut, hindgut, or pancreas. Lanreotide was associated with significantly prolonged progression-free survival [21].

In addition to SSAs, chemotherapy may be used as an alternative treatment option for patients with high-grade NETs. Subsequently, Everolimus and 177Lu-DOTATATE emerged as novel therapeutic options in NETs [19]. Everolimus, an mTOR pathway inhibitor, was evaluated in functioning and non-functioning NETs of the GI tract in The RADIANT-2 and RADIANT-4 trials. Both studies showed improvement of PFS with everolimus compared to placebo. However, it was observed that everolimus effectiveness appears less in midgut NETs (which represented the majority of patients in the RADIANT-2 study) compared with non-midgut NETs [22,23]. The majority of well-differentiated NETs express high levels of somatostatin receptors to which somatostatin analogues bind. Lutetium-177 (177Lu)–Dotatate is a radiolabeled SSA, also known as peptide receptor radiotherapy (PRRT), which targets tumor cells via delivery of systemic radiotherapy. The NETTER-1 trial showed that 177Lu-DOTATATE produced significantly longer PFS and a higher response rate than high-dose octreotide [24].

To identify potential areas for improvement in the diagnosis and management of SB-NET. We aim to report our 10-year experience at The Ottawa Hospital (TOH).

## 2. Materials and Methods

This is an observational retrospective cohort study. We identified SB-NET cases by searching hospital databases using ICD 10 codes. We examined consecutive records of all SB-NET cases referred to/or diagnosed in TOH between 1 January 2011 and 31 December 2021. Only patients with biopsy-proven diagnosis of SB-NET were included in the study. Details of the tumor site, clinical, pathological, management, and outcomes data were recorded. Data collected from the electronic medical records (Epic) were stored on Microsoft® access database software. Results were analyzed using MS Excel and SPSS 25.0 software [25]. We used descriptive statistics to summarize the demographic and clinicopathological data of the patients. We examined categorical variables by utilizing the chi-squared test and Fisher's exact test. For the analysis of survival data (RFS, PFS, OS), we employed the Kaplan–Meier methods and compared the results using the log-rank test. Overall survival (OS) was calculated from the date of tissue diagnosis to the date of death or last follow-up. Recurrence-free survival (RFS) was calculated from the date of surgical intervention to the date of recurrence or death or last follow-up. Progression-free survival (PFS) for SSAs was calculated from the date of starting treatment to the date of confirmed progressive disease or death or last follow-up. Cox proportional hazards model was used to assess the potential prognostic factors. Two-tailed P-values were reported, and a *p*-value of less than 0.05 was considered statistically significant.

## 3. Results

### 3.1. Patient's Characteristics

We identified 177 SB-NET cases diagnosed between 2011–2021. The patient's characteristics are summarized in Table 1. The average age was 66 years and males were 51% (*n* = 91).

**Table 1.** Baseline patient's characteristics.

|  |  | *n* = 177 | (%) |
|---|---|---|---|
| **Age** | Mean | 66 |  |
| **Gender** | Male | 91 | 51% |
|  | Female | 86 | 49% |
| **Comorbidities** | HTN | 74 | 42% |
|  | DM | 22 | 12% |
|  | IHD | 34 | 19% |
| **Clinical presentation** | Diarrhea | 42 | 24% |
|  | Abdominal pain | 81 | 46% |
|  | GI bleeding | 10 | 6% |
|  | Anemia | 13 | 7% |
|  | Bowel obstruction | 23 | 13% |
|  | Carcinoid symptoms | 23 | 13% |
| **Duration of symptoms (pre-diagnosis)** | <14 days | 21 | 12% |
|  | >14 day | 16 | 9% |
|  | >2 months | 21 | 12% |
|  | >6 months | 42 | 24% |
|  | NA | 43 | 24% |
| **Mode of initial diagnosis** | CT | 117 | 66% |
|  | Endoscopy | 26 | 15% |
|  | Surgical exploration | 4 | 2% |
|  | Ultrasound | 3 | 2% |
|  | NA | 24 | 14% |
| **Curative surgery** |  | 120 | 68% |

HTN: Hypertension, DM: Diabetus Mellitus, IHD: Ischemic heart disease, GI: Gastrointestinal, NA: Not available, CT: Computed Tomography.

### 3.2. Sites

The primary sites were ileum 53% (*n* = 94), while duodenum and jejunum represented 9% (*n* = 16) and 7% (*n* = 12) respectively.

### 3.3. Clinical Presentations

Approximately 24% (*n* = 42) had symptoms (not exclusive to carcinoid symptoms) for more than six months before diagnosis, while 12% (*n* = 21) had acute symptoms for less than 14 days. In nearly 29% (*n* = 51) of patients, the diagnosis of SB-NET was an incidental finding during workup for another problem. Twenty-three patients reported carcinoid symptoms at the time of diagnosis mainly flushing 10% (*n* = 18). GI bleeding occurred in 6% (*n* = 10). Thirty-two patients (18%) had functioning NET during the course of the disease.

### 3.4. Pathology

Details of pathology data are summarized in Table 2. The majority of patients had locally advanced or metastatic disease at the time of presentation with stage III, and stage IV representing 42% (*n* = 75) and 41% (*n* = 75), respectively. Among patients with stage IV disease, the most common sites of metastasis were the liver, lymph nodes, and peritoneum. The majority of patients 84% (*n* = 148) had well-differentiated histology, while poorly differentiated histology represented only 1% (*n* = 2). Six patients had high Ki67 (>20%).

**Table 2.** NETs Pathology data.

|  |  | *n* = 177 | (%) |
|---|---|---|---|
| **Site** | Duodenum | 16 | 9% |
|  | Jejunum | 12 | 7% |
|  | Ileum | 94 | 53% |
|  | Small bowel, Unspecified | 55 | 31% |
| **Differentiation** | Well | 148 | 84% |
|  | Moderate | 4 | 2% |
|  | Poor | 2 | 1% |
|  | NA | 23 | 13% |
| **Grade** | 1 | 98 | 54% |
|  | 2 | 48 | 27% |
|  | 3 | 1 | 6% |
|  | NA | 30 | 17% |
| **Ki67** | <20% | 167 | 94% |
|  | ≥20% | 6 | 3% |
|  | NA | 4 | 3% |
| **Functioning** | Functioning | 32 | 18% |
|  | Non-functioning | 113 | 64% |
|  | NA | 32 | 18% |
| **TNM stage** | I | 7 | 4% |
|  | II | 12 | 7% |
|  | III | 75 | 42% |
|  | IV | 73 | 41% |
|  | NA | 10 | 6% |
| **Sites of metasteses (stage IV)** | Liver | 55 |  |
|  | Peritoneum | 36 |  |
|  | Lymph nodes | 20 |  |
|  | Lung | 6 |  |
|  | Bone | 2 |  |

NA: Not available.

### 3.5. Secondary Malignancies

Thirty-four patients (19%) had additional primary malignancies diagnosed either before or after SB-NET. Among them, six were colon cancer and eight were breast cancer.

### 3.6. Management and Outcomes

One hundred twenty patients (68%) underwent surgical resection of their primary tumor, including 28 patients (16%) with limited metastatic disease. A total of 21 patients (18%) had recurrence after curative surgery. A total of 62 patients (35%) received first-line SSAs therapy with octreotide (*n* = 32) or Lanreotide (*n* = 30). After progression on SSAs, seven patients had PRRT as second-line therapy. Four patients had chemotherapy as their first-line treatment, while 7 patients received chemotherapy as second-line or third-line treatment and one patient received 4th line chemotherapy. The chemotherapy protocols used in our cohort are capecitabine-temozolomide (*n* = 1), platinum-etoposide (*n* = 3), FOLFOX (*n* = 3), FOLFIRI (*n* = 2), CAV (*n* = 1), doxorubicin (*n* = 1), and streptozocin/5FU

(*n* = 1). Three patients received everolimus as a second-line treatment and one as third-line treatment.

Three patients with inadequately controlled diarrhea in the setting of carcinoid syndrome despite the use of SSAs were treated with Telotristat.

### 3.7. Metastectomy and Local Therapy

Six patients underwent liver metastectomy, while nine patients received radiofrequency ablation (RFA) as a treatment modality for liver metastases with or without metastectomy, and three patients received TACE (transarterial chemoembolization). Two patients were treated with radiation therapy. One patient underwent HIPEC (hyperthermic intraperitoneal chemotherapy) for limited peritoneal metastases.

### 3.8. Survival

The median follow-up time was 49 months. Death was documented in 24 patients (14%) and 12 patients (7%) were lost to follow-up. The Kaplan–Meier curve for overall survival is shown in Figure 1. The 5-year OS was 100%, 91%, 97%, and 73% for stage I, II, III, and IV respectively. The 5-year OS for the whole study cohort was 86.4%. The 5-year RFS was 100%, 80%, and 72% for stages I, II, and III, respectively (Figure 2). Patients who had carcinoid symptoms at diagnosis had a significantly shorter overall survival time compared to those without carcinoid symptoms (*p* < 0.001, Figure 3). The median PFS for patients treated with SSAs (both octreotide and lanreotide) was 105 months (95% CI 46–163). There was a statistically significant difference in PFS between Octreotide and Lanreotide as first-line treatment, with median PFS of 53 months (95% CI 15–90) and 105 months (95% CI 46–163) for octreotide and lanreotide, respectively (*p* = 0.021, Figure 4). However, there was no statistically significant difference in OS based on first-line SSA (*p* = 0.222, Figure 5). More patients in the octreotide group had carcinoid symptoms at baseline compared with the Lanreotide group (13 vs. 2, *p* = 0.002).

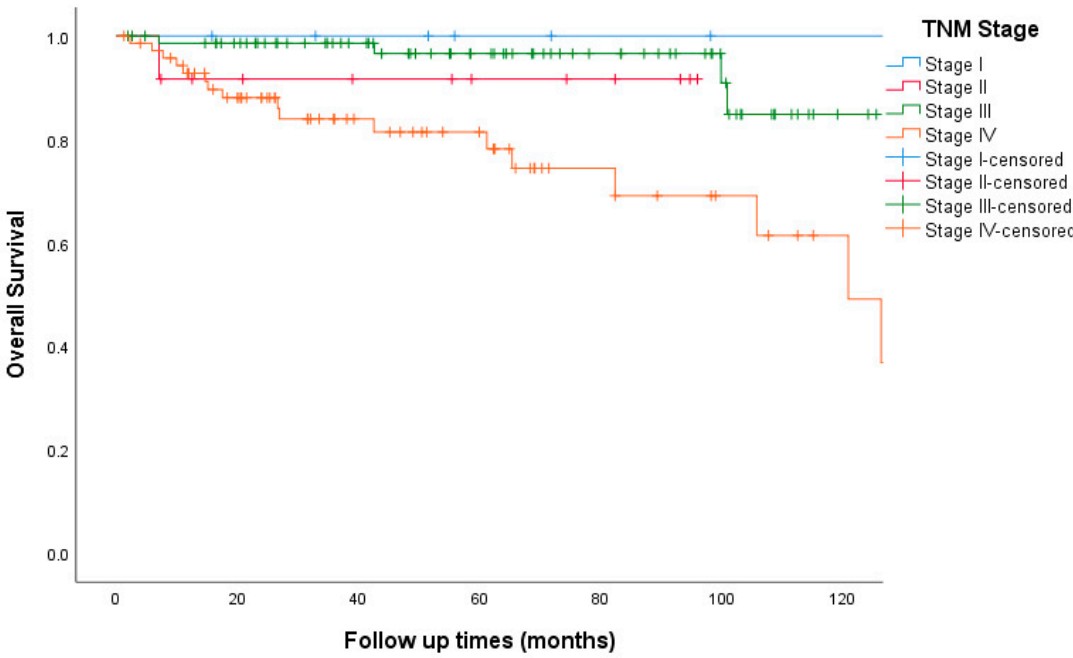

**Figure 1.** Kaplan–Meier curve of overall survival based on TNM Staging.

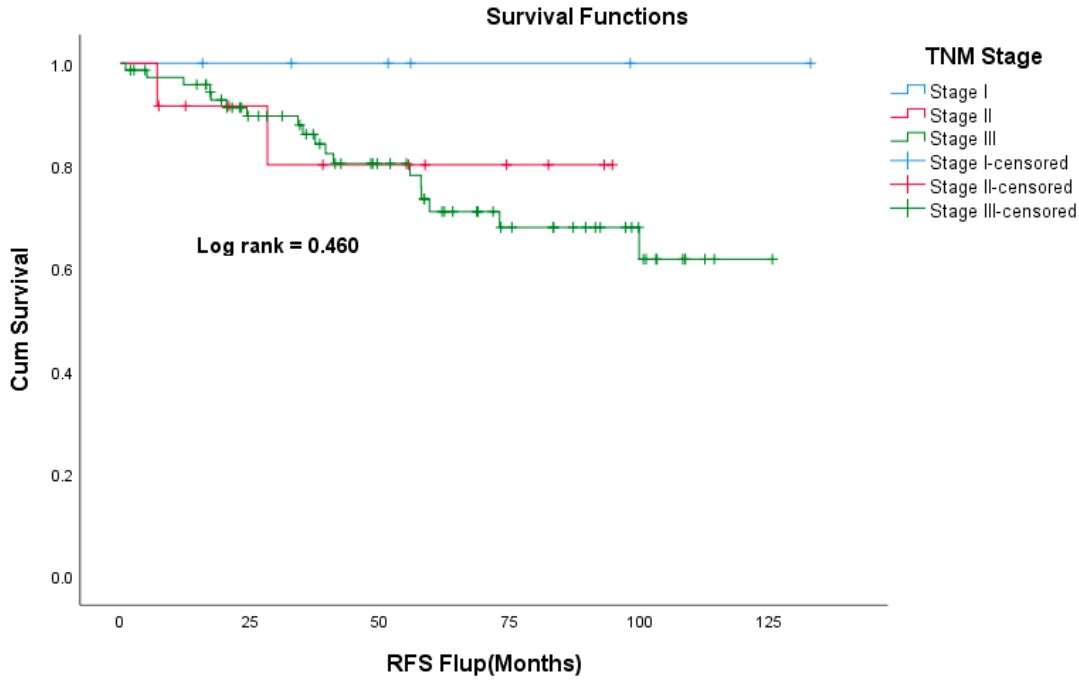

**Figure 2.** Kaplan–Meier curve of recurrence-free survival based on TNM Staging.

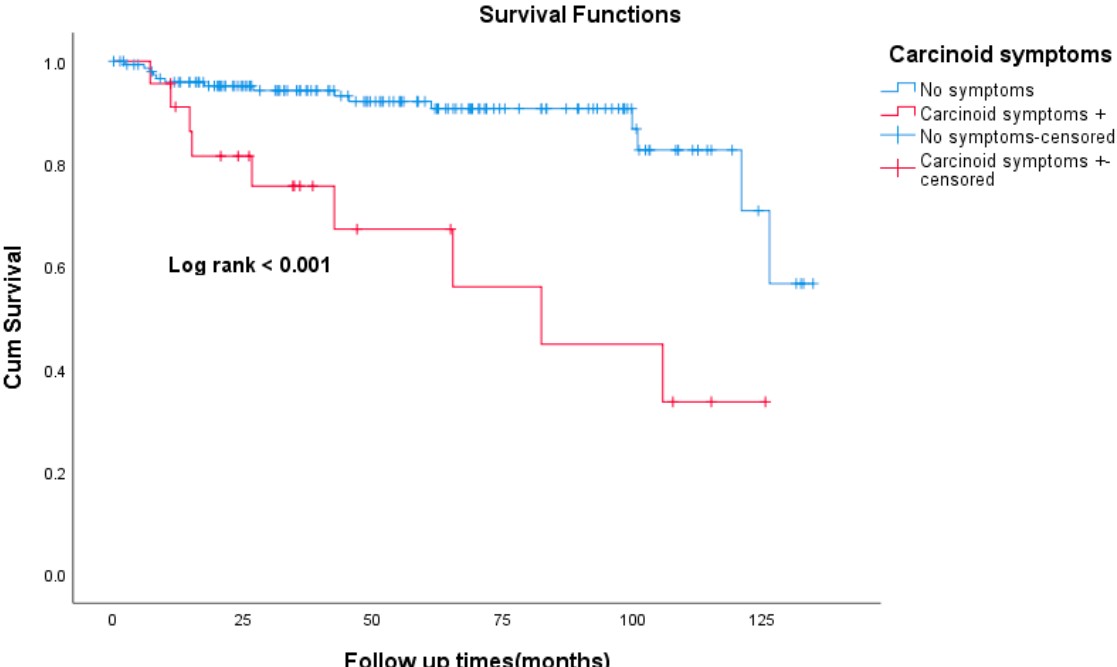

**Figure 3.** Kaplan–Meier curve of overall survival-based presence of carcinoid symptoms.

In univariate analysis, carcinoid symptoms, T stage, and differentiation were significant predictors for OS, but not RFS.

No significant predictors of survival were identified in multivariate analysis, after adjusting for carcinoid symptoms, tumor grade, and TNM stage.

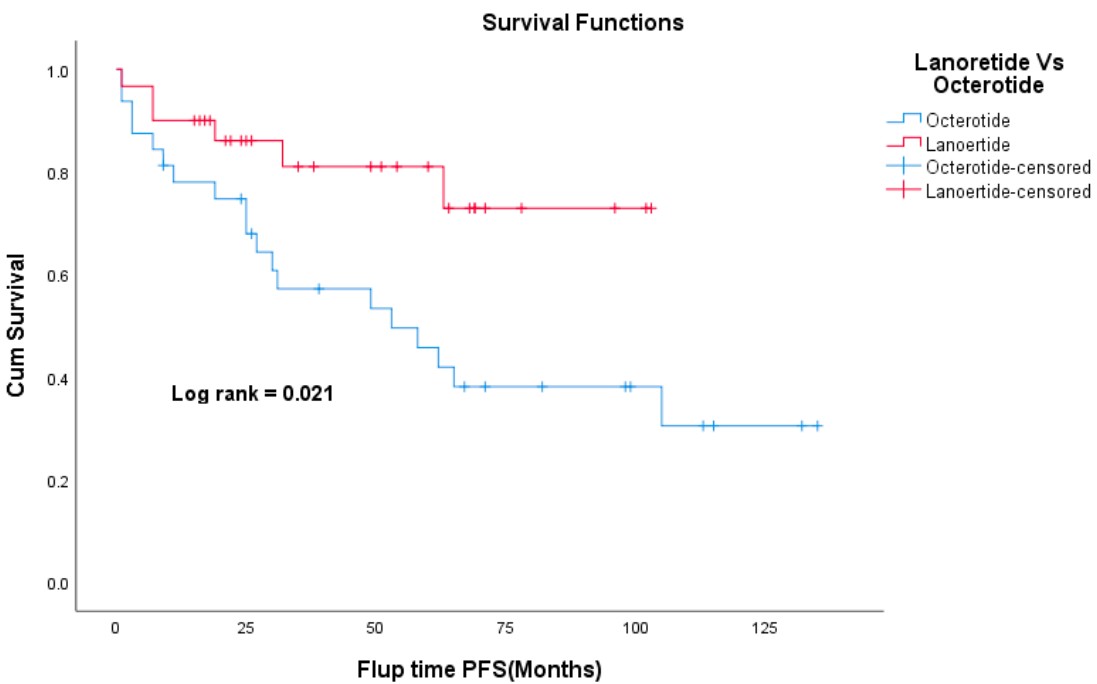

**Figure 4.** Kaplan–Meier curve of progression-free survival for Octerotide and Lanoertide.

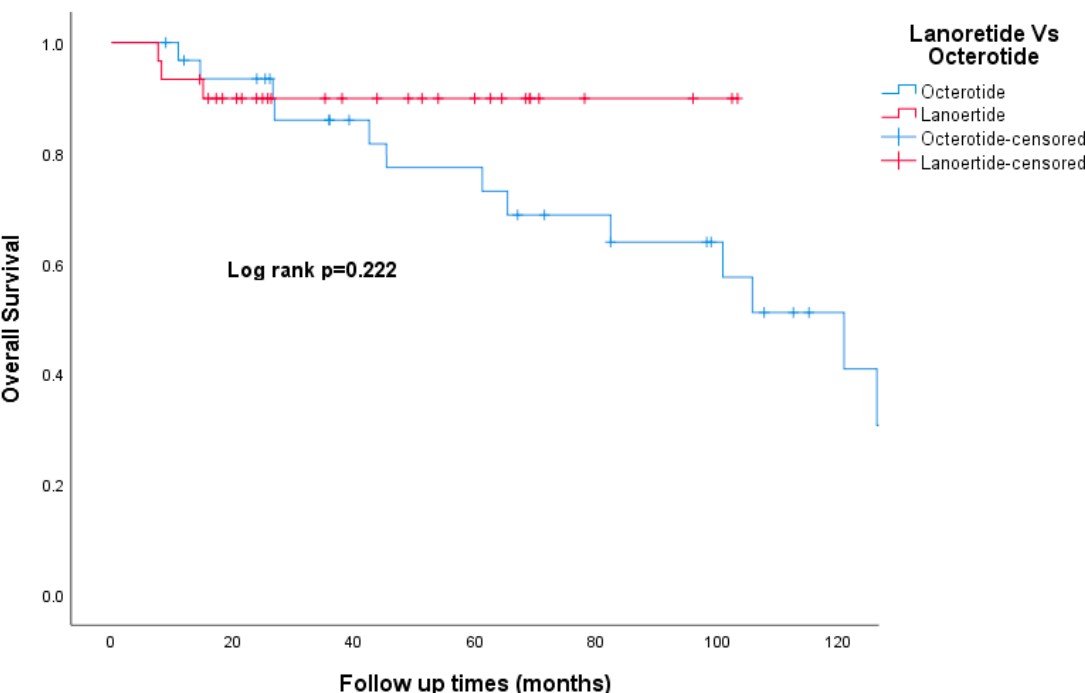

**Figure 5.** Kaplan–Meier curve of overall survival for Octerotide and Lanoertide.

## 4. Discussion

Despite the progress that has been made in the management and understanding of SB-NETs, the survival rates of these patients have not changed over the past three decades [26]. The present study explores the clinicopathological features and outcomes of SB-NET cases seen at the Ottawa Hospital over the past 10 years.

Historically, approximately 70% of patients with SB-NETs have metastatic disease at presentation [27]. In our study, we found that only 41% of patients with SB-NET had stage IV disease at diagnosis. This could reflect the increased usage of cross-sectional imaging

to investigate undiagnosed ailments since 29% of our patients were diagnosed with NET as an incidental finding on imaging. However, it is also difficult to accurately estimate metastases with conventional imaging, and upstaging with a nuclear scan is possible. The use of Ga-DOTATATE PET/CT imaging has improved the diagnosis and staging of patients with NETs compared to conventional imaging [28–30]. In our study, a CT scan was the most useful modality to provide a provisional diagnosis. Given the unavailability of Ga-DOTATATE PET/CT in our institution, few patients had this scan. Data on the effect of access to this scan on patients' outcomes is lacking.

The routine use of molecular diagnostic strategies to guide treatment selection is still limited in real-world practice. However, the development and use of molecular diagnostic strategies, such as 51-gene signature and the detection of circulating tumor cells, have the potential to improve the care of patients with NETs. These strategies can help to identify patients who are more likely to benefit from specific treatments, and they can also be used to monitor the response to treatment and detect recurrence [31,32].

In our study, no patient received adjuvant therapy, as there is currently no evidence to support its use in resected SB-NET tumors. A retrospective study by J. Barrett et al. of 91 patients who received adjuvant therapy either chemotherapy or SSA, found no significant difference in RFS or OS between the two groups [33]. Prospective RCT to investigate the role of adjuvant therapy in this patient population is warranted.

The watch-and-wait strategy could provide an alternative to surgery for some NETs patients such as pancreatic NETs [34]. However, it is less commonly used for SB-NETs than pancreatic NETs because SB-NETs are more likely to be symptomatic at presentation. There are no clear biomarkers that can be used to identify patients who are most likely to benefit from the watch-and-wait strategy. More research is needed to determine the safety and efficacy of the watch-and-wait strategy especially for advanced nonfunctioning SB-NETs.

SSAs are useful to control the symptoms related to NETs and delay their progression. The National Cancer Network (NCCN) has placed both Lanreotide and Octreotide LAR at parity in terms of their overall clinical benefit. Although these two formulations have been used interchangeably, there is not enough evidence to confirm their equivalence. Allaw MB et al. conducted a retrospective study to compare the effects of the two drugs and found there was no difference in PFS between octreotide LAR and Lanreotide for patients with well-differentiated, metastatic GEP-NETs [35]. In our study, there was a statistically significant difference in PFS between octreotide and Lanreotide as first-line treatment. Of note, patients in the octreotide group have more carcinoid symptoms than the Lanreotide group which could bias our results in favor of a less aggressive group of patients. PRRT is now a routine treatment option for patients with SB NET who have failed SSAs. A recently released final analysis of the NETTER-1 trial didn't show statistically significant improvement in OS with a median OS of 48 months in the 177Lu-Dotatate group vs. 36·3 months in the control group [24]. In our study, only seven patients had PRRT as second-line therapy after progression on SSA with a median OS of 126 months; however, selection bias may have contributed to this result.

The role of cytotoxic chemotherapy in patients with advanced NET is controversial largely because of the limited response rate [36]. NCCN guidelines list Fluorouracil, capecitabine, dacarbazine, oxaliplatin, streptozocin, and temozolomide as category 3 recommendations [37]. In our study, chemotherapy use was limited with only four patients receiving chemotherapy as their first-line treatment and 7 patients receiving chemotherapy as second or third-line treatment.

Telotristat ethyl is an oral inhibitor of tryptophan hydroxylase that can effectively reduce the number of bowel movements in patients with refractory carcinoid syndrome diarrhea. A phase III trial (TELESTAR) showed that 44% and 42% of patients treated with 250 mg or 500 mg three times a day had a durable response. In our study, only 3 patients were treated with Telotristat with variable responses.

The survival rate of patients with NETs is influenced by the location of the primary tumor, the presence of distant metastases, and the degree of nodal involvement. A report from a Spanish registry by Garcia-Carbonero and colleagues reported that 5-year survival rates for patients with midgut NETs were 90%, 83%, and 60% for local, regional, and advanced stages respectively [38]. Data from the Surveillance, Epidemiology, and End Results (SEER) program reported the 5-year survival rate to be 69% for SB-NET for the period 2000–2012 [39]. In contrast, data from Canada by Hallet et al. reported 5-year overall survival of 73.4% for (*n* = 1015) patients with SB-NET in Ontario for the period 1994–2009 [40]. Our study found a 5-year survival rate of 86%, which is better than these studies; this finding may be attributable to a higher proportion of well-differentiated diseases in our cohort, in addition to access to SSA.

Immunotherapy has revolutionized the outcomes of different types of cancer, but its role in SB-NETs is still uncertain. There is growing evidence that immunotherapy may be effective for some patients with NETs, but more research is needed to determine the optimal way to use it. Many ongoing clinical trials (e.g., NCT03043664, NCT04525638) are exploring ways to combine immunotherapy with other therapies, such as chemotherapy or SSA, to increase the tumor's susceptibility to immunotherapy [41].

In the era of personalized medicine, there is no clear consensus on the best treatment for SB-NETs because there are no definitive predictive markers and few comparative randomized trials. Treatment decisions should be individualized, taking into account patient factors such as clinical and pathological features.

The limitations of this study are inherent to its retrospective design and include relatively small sample size and short follow-up time. In addition, few patients received second-line therapy (including PRRT) and therefore we are unable to reliably estimate outcomes beyond first-line therapy. Unfortunately, we were unable to include data on 5-HIAA levels in our study, as this important biomarker for carcinoid syndrome was not available for most patients. This was due to issues in retrieving data from the old electronic medical record system. Finally, this study was conducted in one center, so it may not be generalizable to other populations.

The next step for our research is to evaluate the outcomes of patients who received PRRT. We will also evaluate access to gallium PET scans as well as clinical trials on outcomes of SB-NET patients.

## 5. Conclusions

Compared to published historical controls, our study suggests improvement in the 5-year survival rate of small bowel neuroendocrine tumors over the last 10 years.

**Author Contributions:** A.A. (Abdulhameed Alfagih): Conceptual development of the study, literature search, statistical analysis, analysis and interpretation of results, drafting the manuscript. A.A. (Abdulaziz AlJassim), N.A.: data collection, review of the manuscript. M.V., R.G.: conceptual development of the study, critical review of the manuscript. T.A.: conceptual development of the study, supervision, and critical review of the manuscript. All authors have read and agreed to the published version of the manuscript.

**Funding:** This research received no external funding.

**Institutional Review Board Statement:** The study was conducted in accordance with the Declaration of Helsinki, and approved by the Institutional Review Board (or Ethics Committee) of Ottawa Hospital Research Institute (protocol code 20210770-01H and date of approval 3 December 2021).

**Informed Consent Statement:** Not applicable.

**Data Availability Statement:** The data that support the findings of this study are available from the corresponding author upon reasonable request.

**Conflicts of Interest:** The authors have no relevant financial or non-financial interest to disclose.

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
