# Peer review of "Small Bowel Neuroendocrine Tumors—10-Year Experience of the Ottawa Hospital (TOH)"

_curroncol, doi:10.3390/curroncol30080544_

Round 1

Reviewer 1 Report

The authors conducted a retrospective analysis of their cohort of small bowel NETs. However, partially due to he small sample size and the retrospective nature of the study, there are no original and/or new data for the readers, which might potentially affect the clinical practice of NET specialists. As suggested by the author themselves, only few patients received second line treatment, thus I'd suggest the authors to rewrite the paper by adding data regarding other treatments besides SSA.

Meanwhile, the introduction should be implemented. First, it is important to highlight that surgical resection of the primay tumor should be considered also in the metastatic setting (1)       Primary Tumor Resection is Associated with Improved Disease-Specific Mortality in Patients with Stage IV Small Intestinal Neuroendocrine Tumors (NETs): A Comparison of Upfront Surgical Resection Versus a Watch and Wait Strategy in Two Specialist NET Centers Sonja Levy 1James D Arthur 2Melissa Banks 2Niels F M Kok 3Stephen W Fenwick 2Rafael Diaz-Nieto 2Monique E van Leerdam 4Daniel J Cuthbertson 5 6Gerlof D Valk 7Koert F D Kuhlmann # 3Margot E T Tesselaar # 8Br J Surg. 2022 Feb 1;109(2):191-199.  doi: 10.1093/bjs/znab413.

2. Effect of primary tumour resection without curative intent in patients with metastatic neuroendocrine tumours of the small intestine and right colon: meta-analysis Klaas Van Den Heede 1 2 3Swathikan Chidambaram 4 5Sam Van Slycke 3 6 7Nele Brusselaers 8 9Carl Fredrik Warfvinge 2 10Håkan Ohlsson 2 11Erik Nordenström 1 2Martin Almquist 1 2 Eur J Surg Oncol. 2017 Feb;43(2):380-387. doi: 10.1016/j.ejso.2016.10.031. Epub 2016 Nov 24.) . Furthermore, the authors should refer also to the possibility of liver transplant and loco-regional treatment in the introduction. 

In the discussion "This could reflect the increased usage of cross sectional imaging 196 to investigate undiagnosed aliments since 29% of our patients were diagnosed with NET 197 as an incidental finding on imaging. However, it is also difficult to accurately estimate 198 metastases with conventional imaging and upstaging with nuclear scan is possible. The 199 use of Ga-DOTATATE PET/CT imaging has improved the diagnosis and staging of pa- 200 tients with NETs compared to conventional imaging". Plase also consider as a reference: Rossi RE, Massironi S. The Increasing Incidence of Neuroendocrine Neoplasms Worldwide: Current Knowledge and Open Issues. J Clin Med. 2022;11:3794

Finally, the authors should also discuss 5HIAA values in their cohort of patients as this is the most accurate biomarker for carcinoid syndrome.

Author Response

Response to Reviewer 1 Comments

We would like to thank you for your valuable comments and recommendations, which will add to our paper. We appreciate your insights and have taken them into consideration while revising our manuscript.

Point 1:

Adding data regarding other treatments besides SSA

Response 1: Thank you for your suggestion. We understand the importance of providing a comprehensive overview of available treatment options. Our study focused on evaluating the effectiveness of the standard treatment, SSA, as it is recommended as the primary therapy. However, we acknowledge that there are other treatment modalities that have been utilized in clinical practice. we have mentioned some therapies used with or without SSA, in the result section 3.6 and section 3.7, for example, the role of surgery in the metastatic setting, chemotherapy, PRRT, and everolimus.

Point 2:

The role of surgical resection of the primary tumor in the metastatic setting. In addition to role of liver transplants and locoregional therapies.  The reviewer suggested to include 2 studies.

Response 2: We have incorporated your feedback into the introduction section by adding a section about the role of resection of the primary tumor in the metastatic setting. We have also highlighted the role of liver transplants and liver-directed therapies. The suggested references have been included in the revised version. Please see the track changes for more details.

Point 3:

Discussion section: Consider as a reference: R. E. Rossi and S. Massironi, “The Increasing Incidence of Neuroendocrine Neoplasms Worldwide: Cur-rent Knowledge  and Open Issues.,” Journal of clinical medicine, vol. 11, no. 13. Switzerland, Jun. 2022. doi: 10.3390/jcm11133794.

Response 3: Thank you for correctly highlighting this. We have included the suggested references in our revised version ( reference No. 29 )

Please see the track changes.

Point 4:

Discuss 5HIAA values.

Response 4: Thank you for correctly highlighting this. We agree with the reviewer that 5-HIAA levels are an important biomarker for carcinoid syndrome. We would have liked to include this data in our study, but unfortunately, the data was not available for most patients due to issues in retrieving data from the old electronic medical record system.

Reviewer 2 Report

1. The current classification is the WHO classification and not just "based on a consensus meeting in Lyon", I recommend that you rephrase that. 

2. The difference in NET vs. NEC is not only tumor biology, please elaborate. 

3. Which factors did you adjust for in the multivariate analysis?

4. Please remove the grey lines in the Kaplan Maier graphs. 

5. The Discussion is too short. 

Author Response

Response to Reviewer 2 Comments

We would like to thank you for your valuable comments and recommendations, which will add to our paper. We appreciate your insights and have taken them into consideration while revising our manuscript.

Point 1:

The current classification is the WHO classification and not just "based on a consensus meeting in Lyon", I recommend that you rephrase that.

Response 1: Thank you for your suggestion. We have updated the text to reflect the current WHO classification.

The updated text now reads:  The current WHO classification of neuroendocrine neoplasms (NEN), established in a consensus meeting in Lyon, divides them into NETs and neuroendocrine carcinomas (NEC) based on their histological features, growth rate, and molecular differences.

 Please see the track changes.

Point 2:

The difference in NET vs. NEC is not only tumor biology, please elaborate.

Response 2: Thank you for correctly highlighting this. We agree that the difference is not only in tumor biology. There are also clinical and prognostic differences between the two types of tumors.

We have added some additional data to the manuscript to support the reviewer's comment. This data includes:

NECs usually have TP53 or RB1 mutations, while well-differentiated NETs typically have MEN1, DAXX, and ATRX mutations. These distinct genetic profiles contribute to the varying behavior and clinical outcomes observed in NECs and well-differentiated NETs. NECs have a worse prognosis than well-differentiated NETs, and the treatment for NECs is typically more aggressive.

We have added data about the mixed NENs including:

Previously, mixed NENs were classified based on the predominant cell type. However, genomic data have shown that these tumors often have a mixed genetic profile, with fea-tures of both NETs and non-neuroendocrine tumors .As a result, mixed NENs are now grouped into a new conceptual category called "mixed neuroendocrine–non-neuroendocrine neoplasms (MiNENs)". This category includes tumors that have both NET and non-NET features, as well as tumors that have features of both adenocarcinomas and NECs.

Please see track changes.

Point 3:

Which factors did you adjust for in the multivariate analysis?

Response 3: We thank the reviewer for the question about the factors that were adjusted for in the multivariate analysis. We adjusted for the following factors:

  • The presence of carcinoid symptoms
  • The tumor grade
  • The TNM stage

We chose to adjust for these factors because they are known to be prognostic factors for NET. We added a statement in the result section about these factors as following:

No significant predictors of survival were identified in multivariate analysis, after adjusting for carcinoid symptoms, tumor grade, and TNM stage.

 Please see track changes.

Point 4:

Please remove the grey lines in the Kaplan Maier graphs

Response 4: Thank you for correctly highlighting this. These were removed.

 Please see track changes.

Point 5:

The Discussion is too short

Response 5: Thank you for correctly highlighting this. We expanded the discussion section.

 Please see the revised discussion section.

Round 2

Reviewer 1 Report

I do appreciate the efforts made by the authors in addressing all the reviewers' request.

However, I did notice some further points to be addressed:

- in the introducion section re. liver transplant please cite:

1. Mazzaferro V, Pulvirenti A, Coppa J. J Hepatol. 2007 Oct;47(4):460-6.

- In the introduction section, I'd suggest the authors to mention how the diagnosis of small bowel NETs might be challenging (see as a reference United European Gastroenterol J. 2017 Feb;5(1):5-12). Please expalin also in the results section how the diagnosis was made.

- I'd suggest the authors to explain also in the text, as a limiting factor, why data re. 5HIAA were not available for all the authors

Author Response

We would like to thank you for your valuable comments and recommendations, which will add to our paper. We appreciate your insights and have taken them into consideration while revising our manuscript.

Point 1:  in the introducion section re. liver transplant please cite:

  1. Mazzaferro V, Pulvirenti A, Coppa J. J Hepatol. 2007 Oct;47(4):460-6.

Response 1: Thank you for your suggestion. We have added this reference to the text, and you can see the track changes.

Point 2:

- In the introduction section, I'd suggest the authors to mention how the diagnosis of small bowel NETs might be challenging (see as a reference United European Gastroenterol J. 2017 Feb;5(1):5-12). Please expalin also in the results section how the diagnosis was made.

Response 2:

Thank you for your suggestion. We have added a paragraph to the introduction section that discusses the challenges of diagnosing small bowel NETs. The suggested reference was included. We have already explained in the results section how diagnosis was made , please see table 1.

Point 3:

- I'd suggest the authors to explain also in the text, as a limiting factor, why data re. 5HIAA were not available for all the authors

Response 3: Thank you for your suggestion. We have added a paragraph to the limitations section of the manuscript that explains why data on 5-HIAA levels were not available for all patients. Please see tacked changes
